# Technical Note: A simple feedforward artificial neural network for high temporal resolution rain event detection using signal attenuation from commercial microwave links

Erlend Øydvin[1], Maximilian Graf[2, 3], Christian Chwala[2, 3], Mareile Astrid Wolff[1, 5], Nils-Otto Kitterød[4], and Vegard Nilsen[1]

[1]Faculty of Science and Technology, Norwegian University of Life Sciences, Ås, Norway
[2]Institute of Meteorology and Climate Research, Karlsruhe Institute of Technology, Campus Alpin, Garmisch-Partenkirchen, Germany
[3]Institute of Geography, University of Augsburg, Augsburg, Germany
[4]Faculty of Environmental Sciences and Natural Resource Management, Norwegian University of Life Sciences, Ås, Norway
[5]Norwegian Meteorological Institute, Oslo, Norway

**Correspondence:** Erlend Øydvin (erlend.oydvin@nmbu.no)

**Abstract.** Two simple feedforward neural networks (MLPs) are trained to detect rainfall events using signal attenuation from commercial microwave links (CMLs) as predictors and high temporal resolution reference data as target. $MLP_{GA}$ is trained against nearby rain gauges and $MLP_{RA}$ is trained against gauge-adjusted weather radar. Both MLPs were trained on 26 CMLs and tested on 843 CMLs, all located within 5 km of a rain gauge. Our results suggest that these MLPs outperform existing methods, effectively capturing the intermittent behavior of rainfall. This study is the first to use both radar and rain gauges for training and testing for CML rainfall detection. While previous studies have mainly focused on hourly reference data, our findings show that it is possible to classify rainy and dry time steps with higher temporal resolution.

## 1 Introduction

Commercial microwave links (CMLs) are radio links between telecommunication towers. By exploiting the relation between CML signal attenuation and rainfall intensity, it is possible to estimate the average rainfall intensity along the CML (Messer et al., 2006; Leijnse et al., 2007). As the signal is also attenuated by factors other than rain, such as air humidity, these non-rainy factors must be taken into account in what is often called the baseline attenuation. Rain-induced attenuation can then be estimated by subtracting the estimated baseline from the total loss. Since each CML can have a different baseline attenuation, and because the baseline attenuation can change between different rainfall events, it is necessary to estimate the baseline attenuation for each rainfall event. A common approach is to use the signal attenuation from time steps that are temporally close to the rainfall period (Chwala and Kunstmann, 2019; Graf et al., 2020). This raises the need for algorithms that can separate the CML time series into rainy time steps, where the CML experiences signal attenuation due to rainfall, and dry time steps, where the CML signal level is not attenuated by rainfall. This task can be seen as a classification problem, where every time step is classified as either rainy or dry. The separation of the CML time series into rainy and dry time steps can also help

to filter out events in the CML signal time series that show some of the same characteristics as rainfall events but are not caused by rainfall. CML signal loss is recorded differently depending on the network operator and can for instance be available as instantaneous measurements every minute. Another popular format is to record the minimum and maximum signal loss over a period, typically 15 minutes. In this work, we focus on instantaneously sampled CML data as this data is becoming more available, see for instance Andersson et al. (2022) and Covi and Roversi (2024).

The CML signal experiences fluctuations during rain events. Based on this, a simple method for rain event detection was developed by Schleiss and Berne (2010). They suggested using these fluctuations to classify rainy periods by taking the standard deviation of a 60-minute rolling window and setting time steps with values above a certain threshold to rainy. This threshold is different between CMLs, but can be derived from local climate characteristics. Graf et al. (2020) expanded this method by recognizing that climate characteristics are not necessarily valid for different locations, individual years, and in particular specific rainy periods that might be of interest. They proposed to estimate the threshold by computing the 80 % quantile of the 60-minute rolling standard deviation for each CML and multiplying this number by a constant that was found to be similar for all CMLs in the study. A more data-driven approach was explored by Polz et al. (2020). They trained a convolutional neural network (CNN) to detect rainfall events using 800 CMLs in Germany. As a reference, they used the gauge-adjusted radar product RADOLAN-RW from Germany's National Meteorological Service (DWD) which has an hourly resolution. Another approach is to include the signal loss from nearby CMLs (Overeem et al., 2011). This method was shown to work for dense CML networks. The literature describes several other approaches (Habi and Messer, 2018; Reller et al., 2011; Rayitsfeld et al., 2012; Wang et al., 2012).

Although several of the mentioned approaches classify rainfall on a high temporal resolution, all large studies using instantaneous sampled CML data have been evaluated using hourly reference data. This might be a reasonable approach as rainfall detection is mostly used for estimating the baseline, which is typically set as a constant throughout a rainfall event (Chwala and Kunstmann, 2019; Uijlenhoet et al., 2018; Messer and Sendik, 2015). However, existing methods are not optimized for estimating rainfall on a higher temporal resolution, and thus, the estimates might not reflect the true intermittency of rainfall. Estimating too long rainy periods could, in cases where the baseline attenuation drops during the rainfall event, result in a bias where the CML estimates rainfall during time steps where there is no rain. Further, a drawback of estimating too-long rainy periods is that some of the estimated rainy time steps could contain non-liquid precipitation. Because dry snow induces a very low signal attenuation, these time steps appear as dry in the CML time series. Thus, correctly estimating rainy time steps is important because CML time steps that indicate no precipitation could contain dry snow.

In this study, we present two methods for detecting rainy time steps in CML time series data. The goal of both methods is to detect rainy time steps in the time series of a CML where the signal attenuation is provided every 1 minute. This is done with a higher temporal resolution compared to existing methods so that short dry spells during rainy periods can be identified. One method is trained on radar reference data and the other method is trained on rain gauge reference data. Both methods are tested against rain gauge and radar data, highlighting their differences. We also examine the performance of the developed methods in comparison to existing approaches, aiming to gain a clearer understanding of the differences between the two alternative methods.

## 2 Material and Methods

### 2.1 Data

A large dataset with 3901 CMLs from Germany was used, providing transmitted and received signal levels with a temporal resolution of one minute from 01-07-2021 to 31-07-2021. The total signal loss (TL) was computed by subtracting the transmitted signal level from the received signal level. Each CML consists of two-time series called sublinks, reflecting the signal loss in the beams going from location 0 to 1 and vice versa. More information on this dataset can be found in Graf et al. (2020). As ground truth, two different sources were explored. The first used rain gauges near the CMLs provided by DWD. The rain gauge data was provided with a temporal resolution of one minute and a volume resolution of 0.01 mm. We consider a minute to be rainy if the rain gauge records any rainfall. The other source was the radar product *RADKLIM-YW* (Winterrath et al., 2018). This product from DWD is a gauge-adjusted, climatologically corrected product with a temporal resolution of 5 minutes. For the comparison with CML data, the radar product was averaged over the CML path, with each grid value weighted by the length of the CML path intersection in each grid cell. For comparison of the path-averaged *RADKLIM-YW* reference and the CML rainfall estimates, *RADKLIM-YW* was resampled from a 5-minute resolution to a 1-minute resolution by linear interpolation and then dividing the rainfall sums by 5. To make it comparable to the rain gauges, minutes with rainfall above 0.01 mm were set to rainy.

Our study focused on CML-rain gauge pairs located closer to each other than 5 km. This resulted in 882 CMLs where the CML lengths ranged from 0.3km to 22.9km with 90 percent of the CMLs being longer than 2.4km. The CML frequencies ranged between 7 GHz to 40 GHz, with most CMLs having a frequency above 15 GHz. Even though there are many CMLs in our dataset, we only have 429 unique rain gauges serving as references. This means that some CMLs use the same rain gauge for reference.

### 2.2 The $MLP_{RA}$ and $MLP_{GA}$ method

In our approach, we have used a simple feed-forward neural network provided by the python library *sklearn* (Pedregosa et al., 2011). This network consists of an input layer, fully connected hidden layers, and an output layer. Networks with simple architecture of this type are often referred to as a Multilayer perceptron (MLP). The MLPs job is to classify a time step in the CML time series as either rainy or dry. It does this by analyzing the signal loss from the surrounding 40 time steps. In essence, the MLP acts like a sliding window, moving across 40 time steps at a time, and determining whether each centered time step is rainy or dry. The predictor data, that is the 40 time steps moving window, is organized in a so-called design matrix (Equation

1) where $tl_{s_1,t}$ and $tl_{s_2,t}$ represents the total signal loss at time step $t$ for sublink 1 and sublink 2 respectively.

$$
\begin{bmatrix}
tl_{s_1,t_0-20} & \cdots & tl_{s_1,t_0+20} & tl_{s_2,t_0-20} & \cdots & tl_{s_2,t_0+20} \\
\vdots & & \vdots & \vdots & & \vdots \\
tl_{s_1,t_i-20} & \cdots & tl_{s_1,t_i+20} & tl_{s_2,t_i-20} & \cdots & tl_{s_2,t_i+20} \\
\vdots & & \vdots & \vdots & & \vdots \\
tl_{s_1,t_n-20} & \cdots & tl_{s_1,t_n+20} & tl_{s_2,t_n-20} & \cdots & tl_{s_2,t_n+20}
\end{bmatrix}
\tag{1}
$$

We experimented with longer windows, but could not find any improvements by increasing the window size beyond 40 time steps. There was also an improvement from using both sublinks rather than one. This improvement could be because using two sublinks includes more information, which could help the MLP filter out noise.

As pre-processing, we subtracted the 12-hour centered rolling median from the signal level for each CML. This removes longer trends from the signal level making the time series stationary. We experimented with other detrending methods such as differencing, but got poorer results.

Next, two approaches were explored, one where we trained the neural network against radar data ($\mathrm{MLP}_{RA}$) and one where we trained the MLP against rain gauge data ($\mathrm{MLP}_{GA}$). It must be noted that both references observe rainfall at different locations and different spatio-temporal aggregates as compared to the CML. Particularly the rain gauges observe time aggregated point rainfall, whereas the CML observes instantaneous path averaged rainfall. Thus, the references are just an approximation of the rainfall observed by the CML.

For testing, the optimal $\mathrm{MLP}_{RA}$ and $\mathrm{MLP}_{GA}$ were integrated into *pycomlink*, a python library for CML processing (Chwala et al., 2023). Since the current *pycomlink* environment does not support *sklearn*, the weights and network architecture were exported to *tensorflow* using the *Keras* API (Abadi et al., 2015). The final testing was performed by loading the exported MLPs from the pycomlink environment.

## 2.3 Reference methods

Two reference methods were used for comparing the MLP results. The $\sigma_{80}$ method from Graf et al. (2020) and the CNN method from Polz et al. (2020). We note that similar to our MLP, the CNN method is also trained to use two sublinks, whereas the $\sigma_{80}$ method just uses one. Both methods are described in the introduction and can be run from *pycomlink*.

## 2.4 Performance metrics

The performance of the methods was evaluated by recording the classified CML rainy and dry periods against the reference data (rain gauge or radar) in a confusion matrix. In our case, the confusion matrix is a 2x2 matrix listing the number of true positives (TP), true negatives (TN), false positives (FP), and false negatives (FN). Although no perfect performance metric exists, a balanced way of describing the confusion matrix as a single number can be done by the Matthews correlation coefficient (MCC) (Chicco and Jurman, 2020). The MCC is a diagnostic that gives a number between -1 and 1, where 1 represents a perfect classification, 0 is no better than a random guess, and -1 is a perfect disagreement with the reference.

**Table 1.** MLP hyperparameters used in grid search

| Hyperparameter | Values |
| --- | --- |
| Hidden Layer Sizes | [[1], [10], [20], [70], [5, 5], [10, 10], [50, 50], [100, 100]] |
| Activation Function | ['relu', 'logistic'] |
| Regularization | [0., 0.175, 0.35, 0.525, 0.7] |
| Initial Learning Rate | [0.0000001, 0.00000147, 0.00002154, 0.00031623, 0.00464159, 0.06812921, 1] |

## 2.5 Train-test split

In order to assess how well the models performed, the CML data was split into a training set and a test set. Due to, for instance, noisy CMLs, malfunctioning rain gauges, or spatio-temporal uncertainties, some CMLs showed a poor correlation with the rain gauges or the radar. As these pairs could result in poor training data, we opted to exclusively include pairs with high MCC in our training set. We selected training pairs for $MLP_{RA}$ and $MLP_{GA}$ by estimating the CML rainy periods using the $\sigma_{80}$ method. The top 26 CML-radar pairs with the highest MCC, evaluated using radar data as ground truth, were chosen for $MLP_{RA}$. $MLP_{GA}$ used the 26 CML-rain gauge pairs with the highest MCC, evaluated using rain gauge as ground truth. As some of the CMLs share the same neighboring rain gauge, simply selecting the pairs with the highest MCC could make the training data too focused on very similar rainfall events. Thus, to ensure diversity in the training data, the training data used only unique rain gauges. The remaining 843 pairs were used for testing. A possible drawback of this approach is that the MLPs are not trained on noisy CMLs, hindering their effectiveness in dealing with erratic signal fluctuations. However, erratic CMLs are usually removed before the rain event detection step for instance by removing CMLs where the rolling standard deviation of the total loss exceeds 2 dB at least 10% of the time or where the 1 hour rolling standard deviation of the of the total loss exceeds 0.8 dB at least 33% of the time (Graf et al., 2020; Blettner et al., 2023).

## 2.6 Hyperparameter estimation and cross-validation

During training, the MLP classifier can be tuned using several hyperparameters such as activation function, hidden layers, initial learning rate, and L2 regularization. The optimal hyperparameters were found by using k-fold cross-validation over a grid search over the hyperparameter values listed in Table 1. We performed k-fold cross-validation by splitting the CMLs in the training data into 5 folds and iteratively trained the MLP on 4 folds of data and validated on the 5th fold using the MCC. The final score is the mean of all 5 validation MCC scores.

The rainfall time series is characterized by extended periods of no rain, leading to an imbalance that can impede the effectiveness of neural network training. A common method to address this issue is random undersampling, where samples from the majority class are discarded to create a balanced dataset (Hoens and Chawla, 2013). However, rainfall time series often include

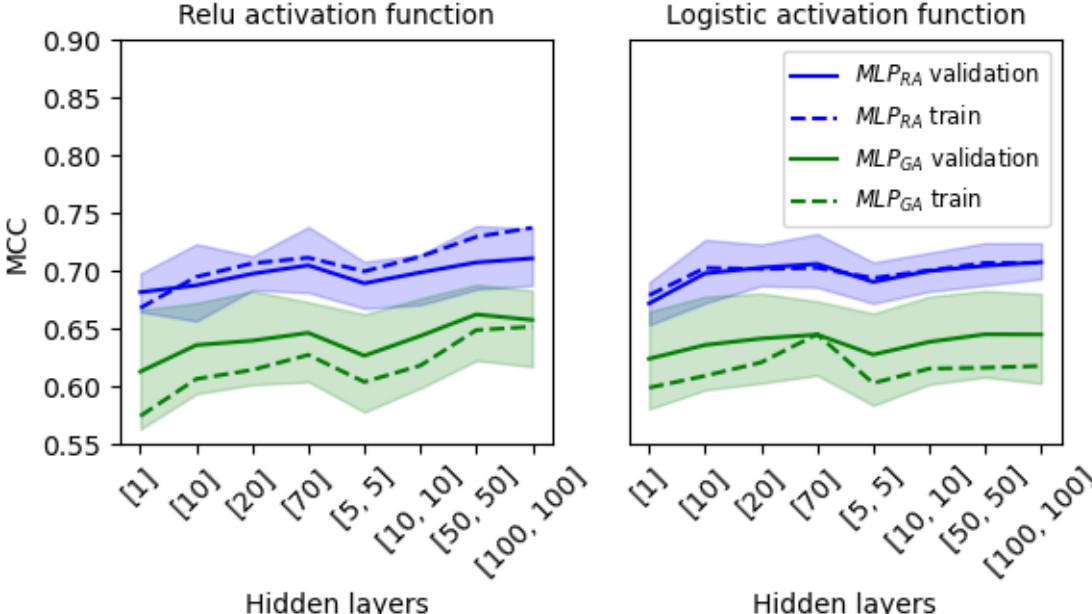

**Figure 1.** MCC as a function of network architecture for the relu and logistic activation function. [5, 5] means two layers with 5 neurons in each layer. The MLP was trained using k-fold cross-validation with 5 folds over 26 CML-rain gauge pairs using radar (MLP$_{RA}$) and rain gauge (MLP$_{GA}$) as reference. The solid line is the mean value of the 5 folds while the shaded area shows the minimum and maximum score of the 5 folds.

short intermittent dry periods within longer events, which are of particular interest in our approach. If we were to use random undersampling, these events might be underrepresented in the training dataset. Recognizing that the total signal loss moving

135    window can include rainy time steps during dry periods close to rainy ones, we have adopted a modified undersampling strategy. Specifically, we only discard dry steps more than 30 minutes away from any rainfall events as detected by the reference methods.

## 3    Results and discussion

### 3.1    Training the MLP

140    The performance (MCC) of MLP$_{RA}$ and MLP$_{GA}$ for the training and test dataset as a function of increased number of neurons and hidden layer sizes is shown in Fig. 1. For each hidden layer configuration, the optimal regularization and initial learning rate that yielded the highest mean MCC was selected and plotted together with the minimum and maximum of all 5 folds obtained from k-fold cross-validation.

**Table 2.** Optimal hyperparameters for the MLP trained with radar reference ($MLP_{RA}$) and the MLP trained with rain gauge reference ($MLP_{GA}$)

| Hyperparameter | $MLP_{RA}$ | $MLP_{GA}$ |
|---|---|---|
| Network architecture | [20] | [50, 50] |
| Activation function | logistic | relu |
| Regularization | 0.175 | 0.175 |
| Initial learning rate | 0.00031623 | 0.00031623 |

We can observe that the $MLP_{GA}$ generally has a lower score than the $MLP_{RA}$ method. This could be because the rain gauges can be located up to 5km away from the CMLs, causing errors related to spatial variability. For the radar data, this issue with spatial representation is most likely mitigated by the comparison based on CML path-weighted intersections. Another reason could be that the spatial averaging performed by the radar and CMLs produces less intermittent rainfall time series than what is the case for the rain gauges, resulting in better agreement between the CML and radar.

The relu activation function has a lower score for simple network architectures (for instance [1]), but produces larger scores with increased network architecture compared to the logistic activation function. Further, for the relu activation function with larger networks ([70] and [100, 100]), $MLP_{RA}$ shows a larger deviation between the train set and validation set, indicating that the model is not generalizing very well. $MLP_{RA}$ has a smaller deviation between train and validation when the logistic activation function is used, indicating more general fits. Thus $MLP_{RA}$ seems to have a good compromise between model complexity and score when using a single layer with 20 neurons and the logistic activation function. $MLP_{GA}$ on the other hand has a smaller deviation between the train and validation set and provides a good compromise between model complexity and score when using two layers with 50 neurons in each and the relu activation function. The optimal hyperparameters for $MLP_{RA}$ and $MLP_{GA}$ are shown in Table 2.

## 3.2 Testing the MLP

The MCC scatter plot density for the $MLP_{RG}$ and $MLP_{RA}$ method compared with the benchmark methods $\sigma_{80}$ and CNN using the radar and rain gauge test data as reference is presented in Fig. 2. For both radar and rain gauge reference we can observe that for most data pairs, the MCC score is higher when using one of the MLP methods than when using one of the reference methods. Another observation is that $MLP_{GA}$ performed slightly better (median MCC of 0.57) than $MLP_{RA}$ (median MCC of 0.52) when the rain gauge was used as a reference. When the radar was used as a reference $MLP_{RA}$ scored slightly better (median MCC of 0.64) than $MLP_{GA}$ (median MCC of 0.60). This difference could be explained by the inherent differences in the measurement methods, where the rain gauge captures the rainfall differently than the weather radar due to for instance wind.

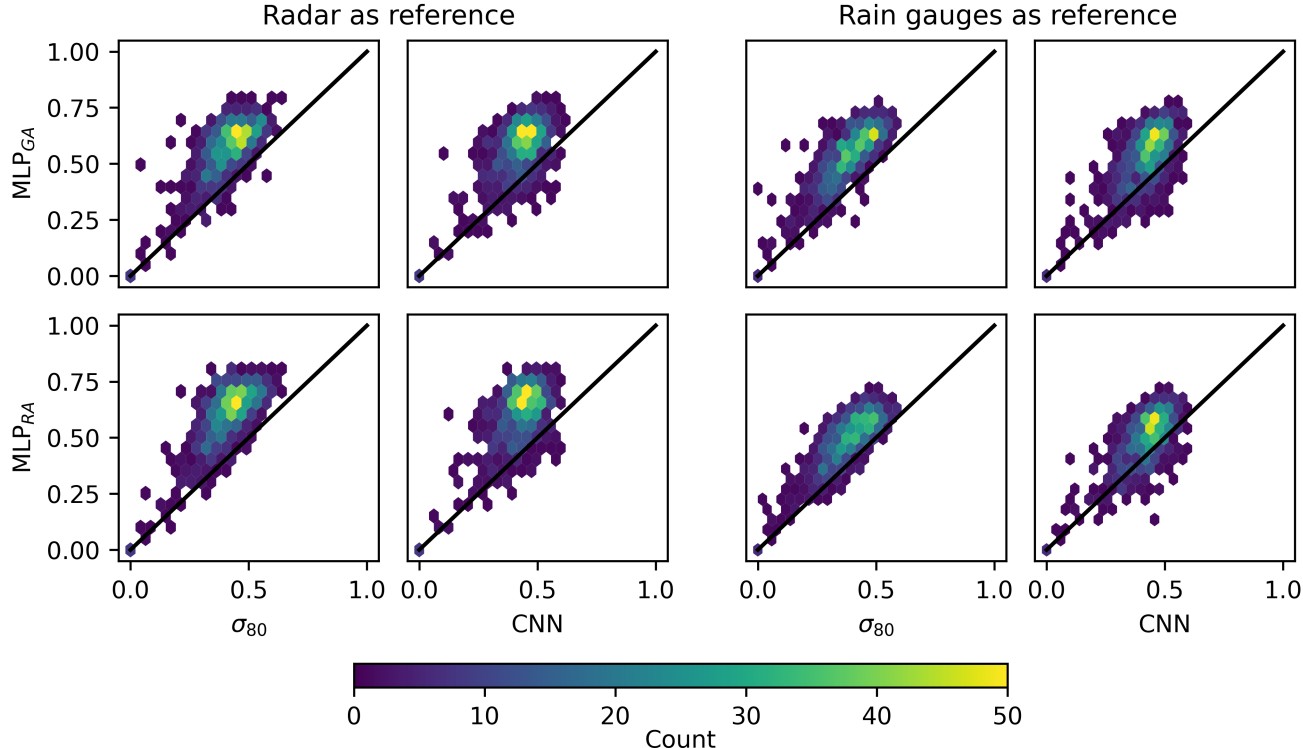

**Figure 2.** Scatter density plot of the MCC score for the MLP trained on the rain gauge reference ($\text{MLP}_{GA}$) and the MLP trained on the radar reference ($\text{MLP}_{RA}$) compared with the benchmark methods $\sigma_{80}$ and CNN. The left plot used radar as reference and the right plot used rain gauges as reference. CML, radar and rain gauge use a one minute resolution. Scores were computed based on 369 CML-radar data pairs over one month.

## 3.3 CML time series

To illustrate how the MLPs perform in comparison to the CNN and $\sigma_{80}$ method, we have selected two events where the MLPs outperform the reference methods (Fig. 3 and Fig. 4) and one event where the MLP performs less well (Fig. 5). The figures show the CML signal loss as a function of time as well as the estimated rainy periods for all methods and the ground truth. We also plot the confusion matrix and the corresponding MCC score for each method using the rain gauge as a reference.

Fig. 3 shows the results from a 10-hour long period for a CML where the $\text{MLP}_{RA}$ method (MCC: 0.73) and $\text{MLP}_{GA}$ method (MCC: 0.76) outperformed the CNN method (MCC: 0.08) and the $\sigma_{80}$ (MCC: 0.47). Looking at the CML total loss (TL) we can observe that the CML has a relatively constant baseline outside the rainy time steps. Around time 06:00 the radar reference (RA) shows a short rainy period, while the rain gauge shows a longer highly intermittent rainy period. The intermittent behavior of the rain gauge might be due to low-intensity rainfall or smaller droplets falling into the scale from the collector. $\text{MLP}_{GA}$ was able to detect a short rainy period at this time whereas $\text{MLP}_{RA}$ did not. For the full 10 hours, the CNN in general estimates

a very long rainy period, missing several dry events and leading to a poorer MCC. This is not surprising as it was trained to detect rainy periods on an hourly basis. The $\sigma_{80}$ method was better in classifying the dry events but still estimated longer rainy periods than the MLPs. Further, $\text{MLP}_{RA}$ tended to estimate rainy periods that started shortly before the CML TL starts to rise, while the $\text{MLP}_{GA}$ tended to estimate rainy periods shortly after the TL has started to rise, see for instance time step 01:00. This is an interesting feature and could be due to the rain gauges showing short breaks at the beginning of rainfall events due to low rainfall intensity. If the beginning of a rainy event has more dry minutes than rainy minutes, as seen by the rain gauge, this could lead $\text{MLP}_{GA}$ to just estimate no rain on these occasions. It could also be caused by radar observing rainfall before it is measured on the ground, making the $\text{MLP}_{RA}$ estimate rainfall shortly before $\text{MLP}_{GA}$.

Fig. 4 shows a 6-hour case for a different CML. Like in Figure 3, $\text{MLP}_{RA}$ estimates a rainy period starting at 12:00, shortly before $\text{MLP}_{GA}$ estimates a wet period. As in the previous case, the CNN estimates a very long rainy period, while the $\sigma_{80}$ estimate rain before and after the rain gauge and radar reference rainfall estimates. In this case, none of the CML rainfall detection methods can accurately estimate the radar or rain gauge reference rainy periods. Looking at the TL we can see that it increases gradually over an extended period, suggesting a longer rainy period. In contrast, the reference data only indicates one or two short rainy events. This discrepancy may be attributed to very low rainfall rates, causing an elevated TL due to CML wet antenna attenuation.

Fig. 3 and Fig. 4 also raise some interesting questions. The final rainfall amount is often derived from a baseline that is typically estimated based on the values of the dry periods before the rainfall event. Since these baseline values are estimated differently for the different methods we have explored in this study, the resulting rainfall rates are expected to vary. For instance, if the $\text{MLP}_{GA}$ is used, the baseline would be placed at a higher level than if the $\text{MLP}_{RA}$ method was used, resulting in a lower rainfall rate estimate. Looking at Figure 3 and the first and last rainfall event detected by $\text{MLP}_{GA}$ (time steps 01.00 and 08.00), it is clear that $\text{MLP}_{GA}$ estimates rainfall shortly *after* the TL has started to rise.

In Figure 5 we have depicted the TL as well as the estimated rainy periods and reference rainy periods for a CML with more erratic signal fluctuations. For $\sigma_{80}$, multiple rainy periods are estimated. While these estimated rainy periods may seem plausible when observing the TL, the reference data reveals that there is no actual rainfall during this time. Therefore, the rainfall estimates likely stem from a noisy CML signal.

### 3.4   General discussion

Our MLPs were trained using CML, weather radar and rain gauge data from 26 CML-rain gauge pairs over one month. The trained MLPs were then tested on 843 CML-rain gauge pairs that were kept out of the training process. A possible limitation of our approach is that one single month might not adequately represent the different rainfall types associated with other months or different geographical locations. On the other hand, since our dataset covers the whole of Germany the dataset contains widely different precipitation events. For instance, in addition to several smaller events, the dataset also captures the large precipitation event that happened in Germany between the 13th and 15 of July 2021. Moreover, to ensure convergence of the MLPs the training data used only 26 CML-rain gauge pairs. Including more pairs, however, did not improve the results on the validation dataset, indicating that the MLPs in fact generalize to several different events.

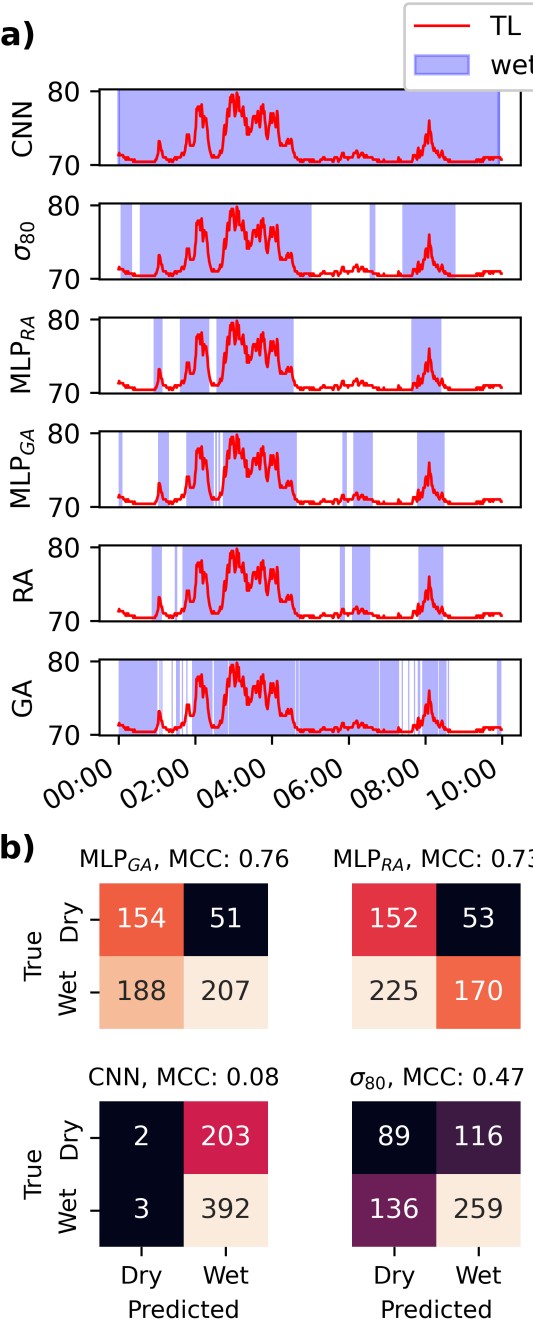

**Figure 3.** a) CML signal loss (TL) for a 10-hour long interval for the CNN, $\sigma_{80}$, MLP$_{RA}$, MLP$_{GA}$ methods. The reference rainy periods for the rain gauge (RG) and gauge-adjusted radar (RA) were also plotted. The blue shaded area marks the rainy periods and the white marks the dry periods. b) Confusion matrix and its corresponding MCC score for the 10-hour period using the CNN, $\sigma_{80}$, MLP$_{RA}$, MLP$_{GA}$ methods with the rain gauge as reference.

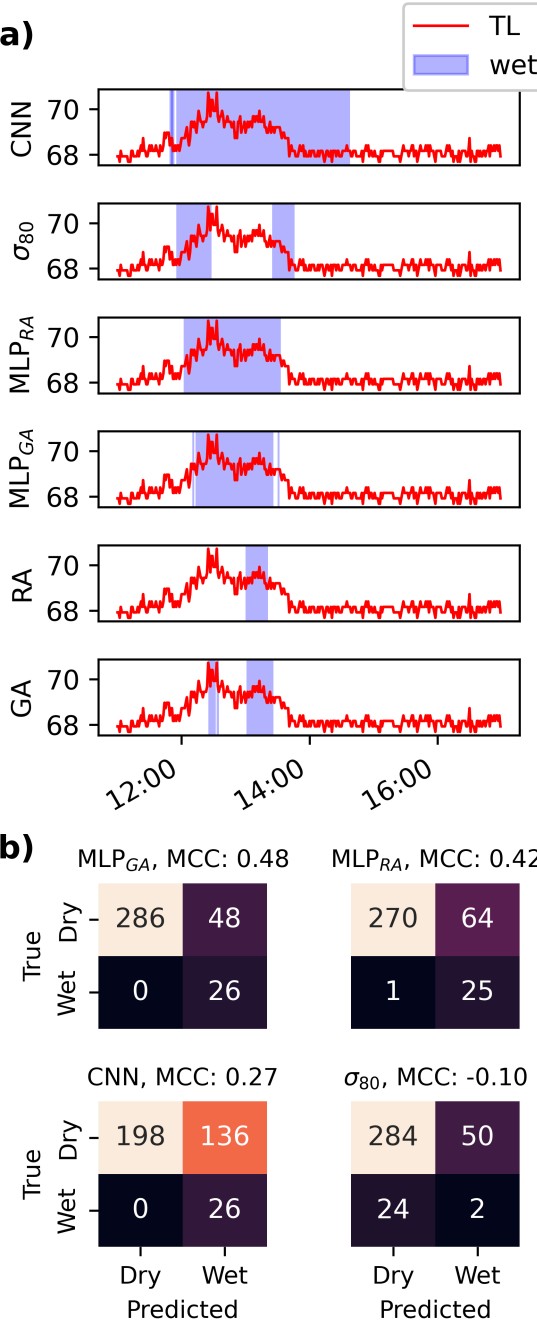

**Figure 4.** a) CML signal loss (TL) for a 6-hour long interval for the CNN, $\sigma_{80}$, $MLP_{RA}$, $MLP_{GA}$ methods. The reference rainy periods for the rain gauge (RG) and gauge-adjusted radar (RA) was also plotted. The blue shaded area marks the rainy periods and the white marks the dry periods. b) Confusion matrix and its corresponding MCC score for the 6-hour period using the CNN, $\sigma_{80}$, $MLP_{RA}$, $MLP_{GA}$ methods with the rain gauge as reference.

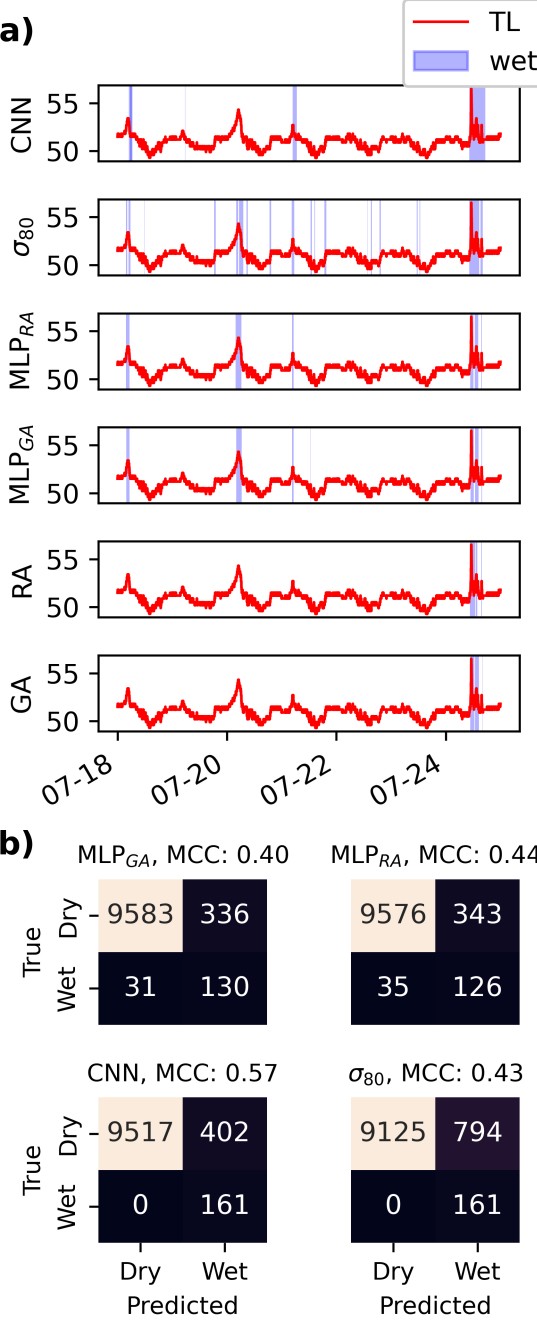

**Figure 5.** a) CML signal loss (TL) for a 6 day long interval for the CNN, $\sigma_{80}$, MLP$_{RA}$, MLP$_{GA}$ methods. The reference rainy periods for the rain gauge (RG) and gauge-adjusted radar (RA) were also plotted. The blue shaded area marks the rainy periods and the white marks the dry periods. b) Confusion matrix and its corresponding MCC score for the 6-day period using the CNN, $\sigma_{80}$, MLP$_{RA}$, MLP$_{GA}$ methods with the rain gauge as reference.

Our results indicate that $MLP_{RA}$ provide rainfall estimates that are more continuous, and more consistent over time, compared to the more intermittent estimates generated by $MLP_{GA}$ (see for instance Fig. 3 time-step 06:00). This could come from the fact that the rain gauges have a 1-minute resolution while the weather radar has a 5-minute resolution, making the radar rainy periods more continuous. Another explanation could be that at low rainfall rates, the rain gauge will not record any rainfall before the droplets have been transported to the scale, making the period seem more intermittent than it actually is. Further, while the rain gauges measure point rainfall close to the CML, the weather radar measures average rainfall along the CML. This path averaging blurs the rainy periods, making the rainy period more continuous with fewer intermittent breaks. An interesting finding is that even though the rain gauges do not represent the average rainfall along the CML, $MLP_{GA}$ is able to capture more of the underlying intermittency as compared to $MLP_{RA}$. This is also reflected in the neural network configuration where the $MLP_{GA}$ benefits from a more complex network architecture as compared to $MLP_{RA}$.

Both MLPs were trained using the 26 CML-reference pairs that showed the highest MCC estimated using the $\sigma_{80}$ method. This can be thought of as a pre-processing step, where the goal was to ensure training data with a good match between the reference and the CML. In our case this was important for making the MLPs converge to approximately the same weights every time we trained the model. These particular pairs might, since they by selection have a good correlation with their reference, also contain little or no noise. Thus, the MLP training datasets might lack exposure to noisy CML time series, and as a consequence, the MLPs might not very well handle noisy periods. On the other hand, from Fig. 2 we know that the MLPs still outperform the $\sigma_{80}$ and CNN method on the 843 CMLs used in the test dataset, which was not subject to any noise filtering, suggesting that the MLPs at least to some extent can handle noise. Moreover, very noisy CMLs are typically handled using pre-processing methods such as filtering out CMLs with strong diurnal cycles or plateaus such as done in Graf et al. (2020) and Blettner et al. (2023).

Overall it must be noted that while the MCC is a useful and balanced metric, its score must be seen in relation to the reference chosen for evaluation. As weather radar provides average rainfall intensities for the entire radar grid cell, we expect that the radar rainfall estimates are less intermittent than what is observed by a rain gauge. This is supported by the findings in Figure 3 where the weather radar rainfall events are less intermittent than what is the case for the rain gauges. The CML, like the weather radar, also measures spatially averaged rainfall. However, the CML measures rainfall closer to the ground and might thus be able to better capture the intermittency as seen by the rain gauge. In this study $MLP_{GA}$ was able to better detect rainfall events as seen by the rain gauge than $MLP_{RA}$. This suggests that there is no single best reference or method for evaluating CML rainy periods. Rather, the CML rain event detection method must be seen in relation to its application.

## 4  Conclusions

In this technical note, we introduced two simple feedforward neural networks (MLPs) trained to detect rainy time steps in signal attenuation data from commercial microwave links (CMLs). The MLPs are trained and tested using reference data from rain gauges ($MLP_{GA}$) with a temporal resolution of 1 minute and gauge-adjusted radar ($MLP_{RA}$) with a temporal resolution of 5 minutes. Whereas existing methods tend to estimate longer continuous rainy periods, the MLPs estimate shorter rainy

periods that more closely resemble the intermittent rainfall patterns that are observed by the rain gauges and weather radar. The performance of the MLPs is evaluated by comparing the MLPs estimates with estimates produced by two existing methods using the Matthews correlation coefficient. Our results show that the MLPs outperform existing methods in almost all cases.

Interestingly, even if the rain gauges do not resemble the path averaged rainfall as observed by the CML, $MLP_{GA}$ was still able to learn the rainfall pattern in the CML time series. Moreover, $MLP_{GA}$ better estimates rainy periods as recorded at the nearby rain gauges than what is the case for $MLP_{RA}$, while both methods perform equally well when radar data is used as reference.

Both MLPs tend to estimate rainy periods after the CML total loss has started to increase. Thus, if the MLPs are used for baseline estimation the user should, similar to Pastorek et al. (2022), consider using dry time steps at least 5 minutes away from the identified rainy time step for baseline estimation.

Future work may involve further refining the model architecture and testing its robustness in generalization to other datasets. Another interesting topic could be to better understand how different wet and dry classifications affect the resulting baselines and the effect this has on rainfall rate estimation from CML data. Overall, both MLPs showed successful skill for the challenge of rainfall event detection in CML attenuation time series.

*Code availability.* The $MLP_{RA}$ method and the $MLP_{GA}$ method are available from pycomlink under https://github.com/pycomlink/pycomlink/tree/master/pycomlink/processing/wet_dry. An example notebook running the different rainfall detection methods is available under https://github.com/pycomlink/pycomlink/tree/master/notebooks

*Data availability.* The rain gauge data was derived from the open data server of the German Meteorological Service and can be found here: https://opendata.dwd.de/climate_environment/CDC/observations_germany/climate/1_minute/precipitation/.

*Author contributions.* Conceptualization: EØ, CC. Data curation: CC. Methodology: EØ, CC, MG, VN, MW, NOK. Software: EØ, MG, CC. Supervision: VN, MW, NOK. Writing – original draft preparation: EØ. Writing – review and editing: EØ, MG, CC, VN, MW, NOK.

*Competing interests.* The contact author has declared that neither of the authors has any competing interests.

*Acknowledgements.* The authors thank co-supervisor Etienne Leblois for nice discussions. The authors would also like to express their gratefulness to the OpenSense COST action (CA20136) for facilitating a short term scientific mission to Garmisch that contributed in making this work possible. We would also like to acknowledge the access to RADOLAN-YW data from the German Weather Service and

270  Ericsson for providing CML data. This work is funded by the Norwegian University of Life Sciences and the German Research Foundation via the SpraiLINK project (Grant CH-1785/2-1).

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
