# Peer review of "Technical Note: A simple feedforward artificial neural network for high temporal resolution rain event detection using signal attenuation from commercial microwave links"

_EGUsphere, 2024_

## Referee Comment (RC1)

**Title**: Review of Manuscript on the Development and Testing of Feedforward Neural Networks for Classifying Wet and Dry Periods Using Commercial Microwave Links

**Reviewer**: Anonymous

Manuscript ID: egusphere-2024-647

Journal: HESS

**Recommendation: Accept with Minor and Major Revisions**

This manuscript presents the development and testing of two simple feedforward neural networks (MLPs) designed for classifying wet and dry periods using signal attenuation from commercial microwave links (CMLs), comparing the performance against existing methods. The focus on high temporal resolution is crucial for enhancing the accuracy of rainfall measurements, representing the manuscript's central novelty. As a technical note, the manuscript sufficiently describes the relevance of the study within the existing literature and highlights the existing gaps it aims to bridge.

**Minor Comments:**

1. The last paragraph of the introduction (paragraph 45) could be more explicit in stating the study's objectives.
2. The first sentence in paragraph 120, subsection 3.1, should be rephrased for enhanced clarity.
3. It would be beneficial for the authors to elaborate on why using total signal loss from both sublinks, as opposed to one, results in improved classification outcomes in paragraph 70.

**Major Comments:**

1. The dataset comprising 3901 CMLs covers only a single month (01-07-2021 to 31-07-2021), which may not adequately represent different rainfall periods and seasonal variations. Expanding the temporal coverage or discussing the potential limitations and implications of this scope on the study's conclusions would strengthen the paper's validity.
2. The manuscript would benefit significantly from a brief discussion regarding the limitations of the study. Such a discussion may include the potential biases introduced by the dataset's temporal limitations (1-minute gauge data versus 5-minute Radar data), the generalizability of the MLP models to other geographic contexts, and the implications of the methodological choices made (e.g., neural network configuration).

---

## Author Comment (AC1)

**Response to reviewer 1:**

We thank the reviewer for their comments and thorough evaluation of our manuscript. The aim of our paper has been to introduce a new method for wet and dry detection that is able to detect wet and dry periods with a higher temporal resolution as compared to existing methods. This allows for more precise identification of rainy time steps, and as a consequence, making identification of short intermittent periods in the CML time series possible. Our main take-away from the two reviewers has been that our paper should have more clearly stated the study's goal and limitations.

Please find our reply in blue to the issues raised.

**Title:** Review of Manuscript on the Development and Testing of Feedforward Neural Networks for Classifying Wet and Dry Periods Using Commercial Microwave Links

**Reviewer:** Anonymous

**Manuscript ID:** egusphere-2024-647

**Journal:** HESS

**Recommendation: Accept with Minor and Major Revisions**
This manuscript presents the development and testing of two simple feedforward neural networks (MLPs) designed for classifying wet and dry periods using signal attenuation from commercial microwave links (CMLs), comparing the performance against existing methods. The focus on high temporal resolution is crucial for enhancing the accuracy of rainfall measurements, representing the manuscript's central novelty. As a technical note, the manuscript sufficiently describes the relevance of the study within the existing literature and highlights the existing gaps it aims to bridge.

**Minor Comments:**

1. The last paragraph of the introduction (paragraph 45) could be more explicit in stating the study's objectives.

We agree that this paragraph can be stated more explicitly. A rephrasing of the paragraph could be "In this study we present two methods for detecting rainy time steps in CML time series data. The goal of both methods is to detect rainy time steps in the time series of a CML where the signal attenuation is provided every 1 minute. This is done with a higher temporal resolution compared to existing methods so that short dry spells during rainy periods can be identified. One method is trained on radar reference data and the other method is trained on rain gauge reference data. Both methods are tested against rain gauge

and radar data, highlighting their difference. We also examine the performance of the developed methods in comparison to existing approaches, aiming to gain a clearer understanding of the differences between the two alternative methods."

2. The first sentence in paragraph 120, subsection 3.1, should be rephrased for enhanced clarity.

We agree with this. The rephrased sentence could be "The performance (MCC) of MLP_RA and MLP_GA for the training and test dataset as a function of the number of neurons and hidden layer sizes is shown in Figure 1. For each hidden layer configuration, the optimal regularization and initial learning rate that yielded the highest mean MCC was selected and plotted together with the minimum and maximum of all 5 folds obtained from k-fold cross-validation.

3. It would be beneficial for the authors to elaborate on why using total signal loss from both sublinks, as opposed to one, results in improved classification outcomes in paragraph 70.

This is indeed an interesting aspect worth elaborating further. However, we do not know exactly why this is the case, but we do know that there is an improvement. Adding too much speculation in the methods section is not desirable, but we could for instance add:

- Paragraph 70: There was also an improvement from using both sublinks rather than one, possibly because two sublinks include more information than one, which could help the MLP filter out noise. Note that the CNN also uses two sublinks whereas the sigma_80 method just uses one. As this topic was not the focus of this study we do not show these findings in this note.
- Paragraph 80: We note that, similar to our MLP, the CNN method is also trained to use two sublinks, whereas the sigma_80 method just uses one.

**Major Comments:**
1. The dataset comprising 3901 CMLs covers only a single month (01-07-2021 to 31-07-2021), which may not adequately represent different rainfall periods and seasonal variations. Expanding the temporal coverage or discussing the potential limitations and implications of this scope on the study's conclusions would strengthen the paper's validity.

It is a good point that our dataset may not fully represent different rainfall periods and seasonal variations. The choice of using a temporally limited dataset was driven by several factors:

a) Since our dataset covers the whole of Germany using 395 CML-rain gauge pairs we believe that the dataset still captures sufficiently different rainfall events. For instance, in addition to several smaller rainfall events the dataset also captures the

large precipitation event that happened in Germany between the 13$^{th}$ and 15$^{th}$ of July 2021.

b) As stated in the paper, in order for the training to converge properly, the training data consisted of the top 26 CML-rain gauge pairs that showed good correlation when the sigma_80 method was used for CML wet dry classification. We could not find any improvement when going beyond 26 pairs. This implies that the training dataset consisting of 26 CML-rain gauge pairs contains a sufficient number of different rainfall events to generalize to the 369 other CML-rain gauge pairs. Thus, we do not expect that expanding the dataset to include several more months of data will impact the results significantly. A very interesting case could be to use data from winter months as this data typically consists of different precipitation types such as dry snow and sleet. However, as the rain gauges and weather radar do not distinguish between snow and rain it would be very hard to train a MLP to detect rain, as there is no way of knowing the ground truth.

We propose to reformulate the discussion and conclusion to highlight the points discussed above.

a) Paragraph 90 delete section 2.5. Add the following to paragraph 60 section 2.1: "Our study focused on CML-rain gauge pairs located closer to each other than 5 km. This resulted in 395 pairs of CMLs and rain gauges spread out across Germany. Even though there are many CMLs in our dataset, we only have 249 unique rain gauges serving as references. This means that some CMLs use the same rain gauge for reference."

b) Paragraph 96: specify that in the training data we used data from 26 unique rain gauges.

c) Paragraph 195: Add a discussion on the implication of only using 1 month of data. "Our study comprises CML, weather radar and rain gauge data from 395 CML-rain gauge pairs over one month. A possible limitation is that one single month might not adequately represent the different rainfall types associated with other months or different geographical locations. On the other hand, since our dataset covers the whole of Germany the dataset contains widely different precipitation events. For instance, in addition to several smaller events, the dataset also captures the large precipitation event that happened in Germany between the 13th and 15 of July 2021. Moreover, in order to ensure convergence of the MLPs the training data used only 26 CML-rain gauge pairs. Including more pairs, however, did not improve the results on the validation dataset, indicating that the MLPs in fact generalize to several different precipitation events."

2. The manuscript would benefit significantly from a brief discussion regarding the limitations of the study. Such a discussion may include the potential biases introduced by the dataset's temporal limitations (1-minute gauge data versus 5-minute Radar data), the generalizability of the MLP models to other geographic contexts, and the implications of the methodological choices made (e.g., neural network configuration).

We suggest to extend the results and discussion chapter with another subsection named general discussion containing the discussion of major comment 1 and the other issues in major comment 2:

[revised manuscript text omitted]

---

## Author Comment (AC2)

**Response to reviewer 2:**

We thank the reviewer for their comments and thorough evaluation of our manuscript. The aim of our paper has been to introduce a new method for wet and dry detection that is able to detect wet and dry periods with a higher temporal resolution as compared to existing methods. This allows for more precise identification of rainy time steps, and as a consequence, making identification of short intermittent periods in the CML time series possible. Our main take-away from the two reviewers has been that our paper should have more clearly stated the study's goal and limitations.

Please find our reply in blue to the issues raised.

Comments to Technical Note: A simple feedforward artificial neural network for high temporal resolution classification of wet and dry periods using signal attenuation from commercial microwave links. by Oydvin et al. (2024)

Application of neural network technic on the attenuated signals of commercial microwave links was introduced to estimate the non-precipitation periods during the intermittent rainfall observed by the gauge based measurements. Data sets with 3901 CMLs for a month were analyzed (trained) with references of one minuets interval rain gauge data and high-temporal 5-minuets interval gauge-adjusted radar products, and statistical differences and temporal characteristics of two products (MPL-GA and MPL-RA) were revealed. Estimating of high-temporal precipitation variabilities using the CMLs might be quite promising technique to produce remote area's precipitation fields with difficulties of ground observation or variation of satellite products at near surface level. MLP would be helpful to translate the CMLs attenuation signals into the duration/amount of precipitation quantitatively. This technical note introduced the performance of MLP comparing with former methods proposed by Graf et al. (2020) and Polz (2020). However, I could not catch up quantitively how better the proposed methods were from the contents. Why is estimating the intermitted precipitation important and which kinds of products are expected to be produced by the MPLs? There are many uncertain technical descriptions to understand the data process. Therefore, the note needs fundamental revision before the formal publication on the HESS.

These are valid remarks. We have attempted answering the concerns in the comments below. One detail that could be stated more clearly is that the study used 395 CMLs for training and testing the MLPs. This is explained in the methods section, but is possible to miss. We suggest adding the following line to the abstract to clarify this "Both MLPs were trained on 26 CMLs and tested on 369 CMLs, located within 5 km of a rain gauge."

General comments

1) "wet and dry periods" is hard to understand. There must be a miss-conversion of concepts between "spatial no-precipitation events identified by CML" and "temporal no-precipitation records by the gauge". In the CMLs, "a dry period" may corresponds to a period without precipitation along one link. However, the authors convert the term to the "dry periods" as multiple no-precipitation periods in a time sequence of point-measured intermittent data. There are no detailed explanation of how the 3901 links data were integrated and aggregated by the MLP? Besides, a gauge measurement could only provide the precipitation intensity over the site without any spatial information of the rain system. "precipitation and non-precipitation periods" in the time sequence are virtual signals, and they are depending on the sensor types and data recording interval (I could not understand "minuets interval 0.01 mm resolution gauge", did you use disdrometer or tipping bucket event recorder?). For instance, non-precipitation periods change depending on one-minuets, hourly, daily, or monthly data even in the same location. Please clarify the concepts of "wet and dry periods" to be understood in the sense of meteorology/hydrology with corrections of title and usage in the contents.

We agree that the motivation for wet-dry classification methods could be better presented. Thus we propose to add to paragraph 10 :

-   "Given that each CML can have a different baseline attenuation, and that the baseline attenuation can change between different rainfall events, it is necessary to estimate the baseline attenuation for each individual rainfall event. A common approach is to use the signal attenuation from time steps that are temporally close to the rainfall period and then assume that the baseline is constant during the rainfall event (chwala_cml_2019). This raises the need for algorithms that can separate the CML time series into rainy time steps, where the CML experiences signal attenuation due to rainfall, and dry time steps, where the CML signal level is not attenuated by rainfall. The separation of the CML time series into rainy and dry time steps also seeks to filter out events in the CML signal time series that show some of the same characteristics as rainfall events, but is not caused by rainfall."

We also recognize that the manuscript switches between the term "wet" and "rainy". We propose to change the occurrences of "wet" to "rainy" throughout the text, which also affects the title of the paper. We therefore suggest the new title to be:

-   Technical Note: A simple feedforward artificial neural network for high temporal resolution rain event detection using signal attenuation from commercial microwave links.

There is indeed a difference between a rainy period as recorded along the path averaged CML and a rainy period as recorded by the rain gauge and this should be stated and discussed in the paper. We agree that this should be more clearly stated and suggest:

- *Adding the following to paragraph 75:* "It must be noted that both types of reference data and the CML time-series record rainfall at different locations and at different spatio-temporal aggregation. Particularly the rain gauges record time aggregated point rainfall, whereas the CML observes instantaneous path averaged rainfall. Thus, the references are just approximations to the rainfall detected in the CML data."
- *And the following to the discussion:* "Our results indicate that MLP_RA provide rainfall estimates that are more continuous, and more consistent over time, compared to the more intermittent estimates generated by MLP_GA. This could come from the fact that the rain gauges have a 1 minute resolution while the weather radar has a 5 minute resolution, making the radar rainy periods more continuous. Another explanation could be that at low rainfall rates, the rain gauge will not record any rainfall before the droplets has been transported to the weight, making the period seem more intermittent than it actually is. Further, while the rain gauges measure point rainfall close to the CML, the weather radar measures average rainfall along the CML. This path averaging blurs the rainy periods, making the rainy period more continuous with less intermittent breaks. An interesting finding is that even if the rain gauge do not represent the average rainfall along the CML, the ground truth is still precise enough so that MLP_GA is able to capture more of the underlying intermittency as compared to MLP_RA. This is also reflected in the neural network configuration where the MLP_GA benefits from a more complex network architecture as compared to MLP_RA.

We agree that the aggregation of the CML signal into the MLP could be more clearly explained. We suggest to change paragraph 65 so that it reads:

- "The purpose of the MLPs  is to classify a time step in the CML time series as either rainy or dry. This is achieved by analyzing the signal loss from the surrounding 40-time steps. In essence, the MLP acts like a sliding window, moving across 40 time steps at a time, and determining whether each centered time step is rainy or dry."

The rain gauge data used in this study are provided by DWD, where the minimum resolution is 0,01mm in the data. In the data availability section we have provided a link to where this data can be obtained. To our knowledge the rain gauges used are of type rain[e] weighing precipitation sensor. More information on this rain gauge can be found here: https://www.lambrecht.net/en/products/precipitation/weighing-precipitation-sensor-rain-e

2) I could not understand what do you want to "classify"? Do you want to distinguish rain or no-rain of instantaneous CML signal, or you want to adjust the CML signals to the precipitation intensity? Also, what does "feedforward" mean? Do you want to propose the

new methods or just demonstrate how the MLP works? Please clarify the target (objects) of this technical note.

In this study we want to distinguish rain from no-rain on a higher temporal resolution than what is done by existing methods. The modifications done to the introduction (in response to comment 1) now describe the purpose more clearly.

To further answer comment 2) we propose to modify paragraph 45 so that the study's goals are more explicitly stated:

- "In this study we present two methods for detecting rainy time steps in CML time series data. The goal of both methods is to detect rainy time steps in the time series of a CML where the signal attenuation is provided every 1 minute. This is done with a higher temporal resolution compared to existing methods so that short dry spells during rainy periods can be identified. One method is trained on radar reference data and the other method is trained on rain gauge reference data. Both methods are tested against rain gauge and radar data, highlighting their difference. We also examine the performance of the developed methods in comparison to existing approaches, aiming to gain a clearer understanding of the distinctions between the different methodologies."

The word feedforward is a very common term in machine learning. We propose to keep it.

3) It is concluded that MLPs performed "better" than existing methods. Some statistics showed differences between your products and sigma80/CNN products, but comparisons on long-term sequence failed to estimate the variations (L184, Fig. 5). Please show the clear reasons of why your work is "better", and explain them in the conclusion. You propose the benefit of higher temporal estimates (L6-7), however it looks like failing filter out the short-term noise in Fig.5. Also, if you proud of the higher temporal product shorter than one hour, it is better to scale up the Figure 3 with one-minuets interval.

The MLPs perform better than existing methods when compared to 1-minute and 5-minute reference data. This is clearly shown in Figure 2.  Figure 3 and 4 shows why this is the case: In short, the sigma80 and the CNN estimates much longer rainy periods, possibly because both of them were developed to predict rainfall using references with hourly resolution. The reason we use the MCC as a performance metric is explained in paragraph 85. The MLPs predict shorter wet periods that better reflect the intermittency of the references that record rainfall with a higher resolution. To better highlight this we suggest modifying parts of the conclusion so that it reads:

- "In this technical note, we introduced two simple feedforward neural networks (MLPs) trained to detect rainy time steps in signal attenuation data from commercial

microwave links (CMLs). The MLPs are trained and tested using reference data from rain gauges (MLP_GA) with a temporal resolution of 1 minute and gauge-adjusted radar (MLP_RA) with a temporal resolution of 5 minutes. Whereas existing methods tend to estimate longer continuous rainy periods, the MLPs estimate shorter rainy periods that more closely resemble the intermittent rainfall patterns that are recorded by the rain gauges and weather radar. The performance of the MLPs are evaluated by comparing the MLPs estimates with estimates produced by two existing methods using Matthews correlation coefficient. Our results show that the MLPs outperform existing methods in almost all cases."

Fig 3) clearly shows that the CNN and sigma80 method estimate longer rainy periods compared to the MLPs. Zooming in to a smaller time interval would leave out important details, like CNN estimating a very long rainy period. For assessing the methods' performance we think Fig 2) is very useful, as plotting the full time series of all links is unfeasible. Fig 3, 4 and 5 provides insight into where the MLPs excel and where they fail. We would prefer to keep Fig 3) as it is because it highlights important details for the discussion.

Fig. 5 shows a time series with a noisy CML. Normally, when deriving rainfall rates from CMLs, noisy CMLs are just completely removed from the dataset. The reason we included this was to show the limitations of the MLP and current rainfall detection method. To better highlight this we propose to:

-   rephrase paragraph 150 to read "In order to better illustrate how the MLPs perform in comparison to the CNN and sigma_80 method, we have selected two events where the MLPs outperforms the reference methods (Fig. 3 and Fig. 4) and one event where the MLP performs less well (Fig. 5). The figures show the CML signal loss as a function of time as well as the estimated rainy periods for all methods and the ground truth. We also plot the confusion matrix and the corresponding MCC score for each method using the rain gauge as a reference."
-   add the following to the discussion: "Both MLPs were trained using the 26 CML-reference pairs that showed the highest MCC estimated using the sigma_80 method. This can be thought of as a pre-processing step, where the goal was to ensure training data with a good match between the reference and the CML. In our case this was important for making the MLPs converge to approximately the same weights every time we trained the model. These particular pairs might, since they by selection have a good correlation with its reference, also contain little or no noise. Thus, the MLP training datasets might lack exposure to noisy CML time series, and as a consequence, the MLPs might not very well handle noisy periods. On the other hand, from Figure 2 we know that the MLPs still outperforms the sigma_80 and CNN method on the 369 CMLs used in the test dataset, which was not subject to any noise filtering, suggesting that the MLPs at least to some extent are able to handle noise. Moreover, very noisy CMLs are typically handled using pre-processing methods such

as filtering out CMLs with strong diurnal cycles or plateaus such as done in (Graf2020) and (blettner_2023).
"

Specific comments

Title: Not clear the meaning of "feedforward", "wet and dry period","

- As the term feedforward is quite well known in machine learning we would like to keep it. We propose to change the title to what is stated in a previous comment.

L9 "signal attenuation" of what?

- To make the sentence clearer we suggest changing it to read: "Commercial microwave links (CMLs) are radio links between telecommunication towers. By exploiting the relation between CML signal attenuation and rainfall intensity, it is possible to estimate the average rainfall intensity along the CML"

L17 more and more "available" mean data are becoming open?

- yes, many of the now open datasets use instantaneous measurements,

L19 What is the "wet period", "Based on this,"?

- We suggest changing the sentence to read: "During rainy time steps, the CML signal loss tends to fluctuate more than during dry time steps. Based on this observation, a simple method for rain ..."

L20 You frequently use "prediction", but this term is for the future estimation. Better to reconsider the usage as "estimate" such as in L175.

- We agree on this and suggest using the word "estimate" throughout the paper.

L32 "hourly reference data" means "hourly precipitation data"? Is this gauge data? No study using disdrometer?

- We agree that is a bit unclear and suggest rephrasing to: "Although several of the mentioned approaches classify rainfall with a high temporal resolution, all large studies using instantaneously sampled CML data have been evaluated using hourly reference data."

L38 What is "rainy time step"? Non-liquid precipitation means solid precipitation such as snow fall?

- We propose to introduce the term rainy time steps earlier in the introduction, see suggestion above.

- Non-liquid precipitation could be snow.

L40-41 I could not understand the meaning.

- We propose to change the sentence to read: "Further, a drawback of estimating too long rainy periods is that some of the estimated rainy time steps could contain non-liquid precipitation. Because dry snow induces a very low signal attenuation, these time steps appear as dry in the CML time series. Thus, correctly estimating rainy time steps is important because CML time steps that indicate no precipitation could contain dry snow."

L43 Object of this note is to describe the methods? Please clarify the object here according to the conclusion.

- We propose to change the objective to read: "In this study we present two methods for detecting rainy time steps in CML time series data. The goal of both methods is to detect rainy time steps in the time series of a CML where the signal attenuation is provided every 1 minute. This is done with a higher temporal resolution compared to existing methods so that short dry spells during rainy periods can be identified. One method is trained on radar reference data and the other method is trained on rain gauge reference data. Both methods are tested against rain gauge and radar data, highlighting their difference. We also examine the performance of the developed methods in comparison to existing approaches, aiming to gain a clearer understanding of the differences between the two alternative methods."

2.1 Data: Please clarify the detailed network structure of the CML, such as general distance, distribution, and what is the "near the CML"? Which kind of rain gauge is used? How the instruments could get 1-minuet/0.01mm resolution? Is this optical sensor? 5 minuets resolution of DWD stans for every 5 minuet interval or 5 minuet average?

- The distance to the nearest rain gauge is explained later in the methods chapter.
- We suggest to add the following sentence to paragraph 60, section 2.1: "Our study focused CML-rain gauge pairs located closer to each other than 5 km, and where the CML was shorter than 5 km. This resulted 395 CMLs where the CML lengths ranged from 0.3km to 5km with 90 percent of the CMLs being longer than 1.6km. The CML frequencies ranged between 20 GHz to 40 GHz. Even though there are many CMLs in our dataset, we only have 249 unique rain gauges serving as references. This means that some CMLs use the same rain gauge for reference."
- The rain gauge data used in this study are provided by DWD, where the minimum resolution is 0,01mm in the data. In the data availability section we have provided a link to where this data can be obtained. To our knowledge the rain gauges used are of type rain[e] weighing precipitation sensor. More information on this rain gauge can be found here:

https://www.lambrecht.net/en/products/precipitation/weighing-precipitation-senso
r-rain-e

L60 "wet" may mean the existence of precipitation record. Then, there must be two kind of "wet periods" such as gauge based and radar based? Both periods were defined by the same threshold as 0.01mm per a minuet?

- We have changed the word "wet" to "rainy" to make it clearer. We have also added an explanation for the distinctions between a rainy and dry period in the introduction as well as some discussions regarding the approximation. See comments above

2.3-2.5 Some of this part are separated only by the paragraph. Please reconsider to combine the sub-section into one section composed by the story only by paragraphs.

- we think the current structure makes it easier for the reader to navigate the different concepts

L81 As the comparison of your methods to two previous method is the key, please explain in detail about two reference methods (Sigma80 and CNN).

- They are already explained in the introduction so we would prefer not to repeat them, as we try to keep this technical note as short as possible.

L112-119 This part describe the study direction, and better to move before. "intermittent rainfall" is your focus, so better to explain what it is (how did you detect).

- Good point! We have made changes to the introduction to better answer this. L112-L119 introduces the undersampling strategy.

L122 Delete "given optimal". Such type existed in may places.

- We suggest deleting the duplicate occurrence of "given optimal".

L126 What is "spatial difference"?

- We suggest to rephrase to: "This could be because the rain gauges can be located up to 5 km away from the CMLs, causing errors related to spatial uncertainty."

L131 You insist "more consistently" from which part of the figure? Need more polite explanation to the readers.

- We propose to delete the sentence as the following sentences fully explain the findings.

L150- I could not understand the process making Fig. 3-5. Are they case studies in different time series, or some kind of composite with different time scale? Better to divide upper and

lower part as Fig. 3a and 3b, but the lower parts are not explained fully in the contents. White areas are dry periods?

- Figures 3-5 can be thought of as case studies. We suggest changing paragraph 150 to read "In order to illustrate how the MLPs perform in comparison to the CNN and sigma_80 method, we have selected two cases where the MLPs outperforms the reference methods (Fig. 3 and Fig. 4) and one case where the MLP performs less good (Fig. 5). The figures show the CML signal loss as a function of time as well as the estimated rainy periods for all methods and the ground truth. We also plot the confusion matrix and the corresponding MCC score for each method using the rain gauge as a reference."
- We do not think that adding a) and b) toFigures 3-5 makes them easier to navigate as it should be evident for the reader what is the time series and what is the confusion matrix.
- CML signal loss (TL) for a 6-hour long interval and its corresponding confusion matrix (compared to rain gauge reference) and MCC score for the CNN, sigma_80, MLP_RA, MLP_GA methods. The reference rainy periods for the rain gauge (RG) and gauge-adjusted radar (RA) was also plotted. The blue shaded area marks the rainy periods and white marks dry periods.
- We also propose to change the term "wet" in the legends to "rainy".

L155 Why "nicely"? I can not catch up which part of the figure could correspond to your results. Same asn L161 "Further ,,, rise", etc.

- We suggest to rephrase the sentence to read: "has a relatively constant baseline outside the rainy time steps".
- To better highlight where the tl rises we suggest to add: "see for instance time step 01:00".

L161 "Further, ,,rise." I could not identify those tendency in Fig. 3. Need marks or specify with times.

- To better highlight where the tl rises we suggest to add: "see for instance time step 01:00".

Fig.3 upper: Clear characteristics on Fig.3, that I could identify, were such as 1) CNN is similar to GA and Sigma80 is similar to RA, 2) MPL-GA and MPL-RA estimated longer no-precipitation periods as RA, 3) MPL-GA and MPL-RA looks mostly similar with a small difference around 6:00. If the MLP studied the RA and GA observations respectively, why the MLP-GA and MLP-RA are so similar? Your focuses may be on more small scale, but I could not figure out the importance of your argument. The graph of GA should put above the RA to adjust the order of MPL-GA and MPL-RA.

- Fig3 shows a test and thus we do not expect that MLP_GA should necessarily estimate the same wet periods as observed by the rain gauge. We do expect MLP_GA to learn the signal loss pattern of the CML that best explains the rainfall pattern at the rain gauge and the MLP_RA learns the signal loss pattern of the CML that best explains the rainfall patterns observed by the weather radar. The reason we included both the MLP_RA and MLP_GA was because they seem to estimate slightly different rainy periods compared to each other. Both outperforms existing methods.
- We agree to change the order of MLP-GA and MLP-RA as that makes the comparison more logical.

L168 I could not identify the wet starting point.

- We suggest rephrasing to: "estimates a rainy period starting at 12:00, shortly before MLP_GA estimates a wet period."

L170 I could not understand the description of "non of the methods ,,". What is the "reference period"?

- We suggest rephrasing to: "In this case, none of the CML rainfall detection methods can accurately estimate the radar or rain gauge reference rainy periods."

L173 "However, " I could not understand what you mean.

- We suggest deleting the sentence as it is fully covered by the previous sentences.

L178-182 This part should move to the paragraph in L166.

- We suggest to keep this part as it is and rephrase the paragraphs in L166 to better guide the reader in reading through the figures using time marks such as "estimates a rainy period starting at 12:00, shortly before MLP_GA estimates a wet period."

Fig.5 This is the result of longest record with few precipitation periods in RA GA where MPL reproduced some intermittent periods of rains similar to sigma80. The author mentioned "erratic signals" in L184, and attributed by a noisy CML signal. It means the MLP even could not filter out one noise on the CML? Then why you can conclude "better (L3)" performance?

- From Figure 2 we can see that the MLPs outperform the existing methods. Please note that focusing on these specific time steps of Figure 3-5 was intentional and that by choosing other time steps we would get different results. This particular case was chosen because it shows an edge case where the MLPs fail and we believe that the reader should be aware of this.
- We acknowledge that part of the job of the MLPs is to filter out noise in the CML time series. However, this CML in particular is very noisy and would most likely be disregarded by quality control routines.

- Future developments into wet and dry detection could definitely seek to use for instance deep learning methods to identify fluctuations due to other-than-rain factors in CML time series. This is however far from the scope of this work where we have used very simple network architectures.
- We suggest to address this by adding the following:
  - To paragraph 150: In order to illustrate how the MLPs perform in comparison to the CNN and sigma_80 method, we have selected two events where the MLPs outperforms the reference methods (Fig. 3 and Fig. 4) and one event where the MLP performs less good (Fig. 5). The figures show the CML signal loss as a function of time as well as the estimated rainy periods for all methods and the ground truth. We also plot the confusion matrix and the corresponding MCC score for each method using the rain gauge as a reference.

L190-195 Future issues should be mentioned in the conclusion.

- We think that L190-195 fits best into the discussion part.

---

## Author Response (AR1)

**Authors' response**

We thank the reviewers for their comments and thorough evaluation of our manuscript. The aim of our paper is to introduce a new method for wet and dry detection that can detect wet and dry periods with a higher temporal resolution than existing methods. This allows for more precise identification of rainy time steps and, as a consequence, makes identification of short intermittent periods in the CML time series possible. Our main take-away from the two reviews has been that our paper should have more clearly stated the study's goal and limitations.

The manuscript has undergone several changes based on the reviewers' comments. A major issue we have addressed is a better discussion of the paper's possible limitations. Another issue, that was not explicitly mentioned by the reviewers, was that in the previous version we restricted the study to only using CMLs shorter than 5km. In this updated version, we have relaxed this requirement to include CMLs of any length within 5km of a rain gauge. This resulted in a much larger test dataset, but the results were not significantly changed.

Please find our response in blue to the issues raised.

**Response to reviewer 1:**

**Title:** Review of Manuscript on the Development and Testing of Feedforward Neural Networks for Classifying Wet and Dry Periods Using Commercial Microwave Links

**Reviewer:** Anonymous

**Manuscript ID:** egusphere-2024-647

**Journal:** HESS

**Recommendation: Accept with Minor and Major Revisions**
This manuscript presents the development and testing of two simple feedforward neural networks (MLPs) designed for classifying wet and dry periods using signal attenuation from commercial microwave links (CMLs), comparing the performance against existing methods. The focus on high temporal resolution is crucial for enhancing the accuracy of rainfall measurements, representing the manuscript's central novelty. As a technical note, the manuscript sufficiently describes the relevance of the study within the existing literature and highlights the existing gaps it aims to bridge.

**Minor Comments:**

1. The last paragraph of the introduction (paragraph 45) could be more explicit in stating the study's objectives.

We agree that this paragraph can be stated more explicitly. We have rephrased the paragraph to be: "In this study, we present two methods for detecting rainy time steps in CML time series data. The goal of both methods is to detect rainy time steps in the time series of a CML where the signal attenuation is provided every 1 minute. This is done with a higher temporal resolution compared to existing methods so that short dry spells during rainy periods can be identified. One method is trained on radar reference data and the other method is trained on rain gauge reference data. Both methods are tested against rain gauge and radar data, highlighting their differences. We also examine the performance of the developed methods in comparison to existing approaches, aiming to gain a clearer understanding of the differences between the two alternative methods."

2. The first sentence in paragraph 120, subsection 3.1, should be rephrased for enhanced clarity.

We agree with this and have rephrased the sentence to be "The performance (MCC) of MLP_RA and MLP_GA for the training and test dataset as a function of increased number of neurons and hidden layer sizes is shown in Fig. 1 For each hidden layer configuration, the optimal regularization and initial learning rate that yielded the highest mean MCC was selected and plotted together with the minimum and maximum of all 5 folds obtained from k-fold cross-validation."

3. It would be beneficial for the authors to elaborate on why using total signal loss from both sublinks, as opposed to one, results in improved classification outcomes in paragraph 70.

This is indeed an interesting aspect worth elaborating further. However, we do not know exactly why this is the case, but we do know that there is an improvement. Adding too much speculation in the methods section is not desirable. We have added this:

- Paragraph 70: "There was also an improvement from using both sublinks rather than one. This improvement could be because using two sublinks includes more information, which could help the MLP filter out noise."
- Paragraph 80: We note that, similar to our MLP, the CNN method is also trained to use two sublinks, whereas the sigma_80 method just uses one.

**Major Comments:**
1. The dataset comprising 3901 CMLs covers only a single month (01-07-2021 to 31-07-2021), which may not adequately represent different rainfall periods and seasonal variations. Expanding the temporal coverage or discussing the potential

limitations and implications of this scope on the study's conclusions would strengthen the paper's validity.

It is a good point that our dataset may not fully represent different rainfall periods and seasonal variations. The choice of using a temporally limited dataset was driven by several factors:

a) Since our dataset covers the whole of Germany using 395 CML-rain gauge pairs we believe that the dataset still captures sufficiently different rainfall events. For instance, in addition to several smaller rainfall events the dataset also captures the large precipitation event that happened in Germany between the 13$^{th}$ and 15$^{th}$ of July 2021.

b) As stated in the paper, in order for the training to converge properly, the training data consisted of the top 26 CML-rain gauge pairs that showed good correlation when the sigma_80 method was used for CML wet dry classification. We could not find any improvement when going beyond 26 pairs. This implies that the training dataset consisting of 26 CML-rain gauge pairs contains a sufficient number of different rainfall events to generalize to the 369 other CML-rain gauge pairs. Thus, we do not expect that expanding the dataset to include several more months of data will impact the results significantly. A very interesting case could be to use data from winter months as this data typically consists of different precipitation types such as dry snow and sleet. However, as the rain gauges and weather radar do not distinguish between snow and rain it would be very hard to train a MLP to detect rain, as there is no way of knowing the ground truth.

c) In the updated manuscript we have included CMLs of any length, raising the number of CML-rain gauge pairs to 843.

We have reformulated the discussion and conclusion to highlight the points discussed above.

a) Paragraph 90 delete section 2.5.  Add the following to paragraph 60 section 2.1: "Our study focused on CML-rain gauge pairs located closer to each other than 5 km. This resulted in 882 CMLs where the CML lengths ranged from 0.3km to 22.9km with 90 percent of the CMLs being longer than 2.4km. The CML frequencies ranged between 7 GHz to 40 GHz, with most CMLs having a frequency above 15 GHz. Even though there are many CMLs in our dataset, we only have 429 unique rain gauges serving as references. This means that some CMLs use the same rain gauge for reference."

b) Paragraph 96: specify that in the training data we used data from 26 unique rain gauges.

c) Paragraph 195: Add a discussion on the implication of only using 1 month of data. "Our MLPs were trained using CML, weather radar and rain gauge data from 26 CML-rain gauge pairs over one month. The trained MLPs were then tested on 843 CML-rain gauge pairs that were kept out of the training process. A possible limitation of our approach is that one single month might not adequately represent the

different rainfall types associated with other months or different geographical locations. On the other hand, since our dataset covers the whole of Germany the dataset contains widely different precipitation events. For instance, in addition to several smaller events, the dataset also captures the large precipitation event that happened in Germany between the 13th and 15 of July 2021. Moreover, to ensure convergence of the MLPs the training data used only 26 CML-rain gauge pairs. Including more pairs, however, did not improve the results on the validation dataset, indicating that the MLPs in fact generalize to several different events."

2. The manuscript would benefit significantly from a brief discussion regarding the limitations of the study. Such a discussion may include the potential biases introduced by the dataset's temporal limitations (1-minute gauge data versus 5-minute Radar data), the generalizability of the MLP models to other geographic contexts, and the implications of the methodological choices made (e.g., neural network configuration).

We have extended the results and discussion chapter with another subsection named general discussion containing the discussion of major comment 1 and the other issues in major comment 2:

[revised manuscript text omitted]

Application of neural network technic on the attenuated signals of commercial microwave links was introduced to estimate the non-precipitation periods during the intermittent rainfall observed by the gauge based measurements. Data sets with 3901 CMLs for a month were analyzed (trained) with references of one minuets interval rain gauge data and high-temporal 5-minuets interval gauge-adjusted radar products, and statistical differences and temporal characteristics of two products (MPL-GA and MPL-RA) were revealed. Estimating of high-temporal precipitation variabilities using the CMLs might be quite promising technique to produce remote area's precipitation fields with difficulties of ground observation or variation of satellite products at near surface level. MLP would be helpful to translate the CMLs attenuation signals into the duration/amount of precipitation quantitatively. This technical note introduced the performance of MLP comparing with former methods proposed by Graf et al. (2020) and Polz (2020). However, I could not catch up quantitively how better the proposed methods were from the contents. Why is estimating the intermitted precipitation important and which kinds of products are expected to be produced by the MPLs? There are many uncertain technical descriptions to understand the data process. Therefore, the note needs fundamental revision before the formal publication on the HESS.

These are valid remarks. We have attempted answering the concerns in the comments below. One detail that could be stated more clearly is that the study used 395 CML-rain gauge pairs for training and testing the MLPs. In the revised version we have extended the test dataset to 843 CML-rain gauge pairs. This is explained in the methods section, but is possible to miss. We have added the following line to the abstract to clarify this "Both MLPs were trained on 26 CMLs and tested on 843 CMLs, all located within 5 km of a rain gauge."

General comments

1) "wet and dry periods" is hard to understand. There must be a miss-conversion of concepts between "spatial no-precipitation events identified by CML" and "temporal no-precipitation records by the gauge". In the CMLs, "a dry period" may corresponds to a period without precipitation along one link. However, the authors convert the term to the "dry periods" as multiple no-precipitation periods in a time sequence of point-measured intermittent data. There are no detailed explanation of how the 3901 links data were integrated and aggregated by the MLP? Besides, a gauge measurement could only provide the precipitation intensity over the site without any spatial information of the rain system. "precipitation and non-precipitation periods" in the time sequence are virtual signals, and they are depending on the sensor types and data recording interval (I could not understand "minuets interval 0.01 mm resolution gauge", did you use disdrometer or tipping bucket event recorder?). For instance, non-precipitation periods change depending on one-minuets, hourly, daily, or monthly data even in the same location. Please clarify the concepts of "wet and dry periods" to be understood in the sense of meteorology/hydrology with corrections of title and usage in the contents.

We agree that the motivation for wet-dry classification methods could be better presented. Thus we have added to paragraph 10 :

- "Since each CML can have a different baseline attenuation, and because the baseline attenuation can change between different rainfall events, it is necessary to estimate the baseline attenuation for each rainfall event. A common approach is to use the signal attenuation from time steps that are temporally close to the rainfall period (chwala_2019, Graf2020. This raises the need for algorithms that can separate the CML time series into rainy time steps, where the CML experiences signal attenuation due to rainfall, and dry time steps, where the CML signal level is not attenuated by rainfall. This task can be seen as a classification problem, where every time step is classified as either rainy or dry. The separation of the CML time series into rainy and dry time steps can also help to filter out events in the CML signal time series that show some of the same characteristics as rainfall events but are not caused by rainfall."

We also recognize that the manuscript switches between the term "wet" and "rainy". We propose to change the occurrences of "wet" to "rainy" throughout the text, which also affects the title of the paper. We therefore suggest the new title to be:

- Technical Note: A simple feedforward artificial neural network for high temporal resolution rain event detection using signal attenuation from commercial microwave links.

There is indeed a difference between a rainy period as recorded along the path averaged CML and a rainy period as recorded by the rain gauge and this should be stated and discussed in the paper. We agree that this should be more clearly stated and have added:

- *To paragraph 75:* "It must be noted that both references observe rainfall at different locations and different spatio-temporal aggregates as compared to the CML. Particularly the rain gauges observe time aggregated point rainfall, whereas the CML observes instantaneous path averaged rainfall. Thus, the references are just an approximation of the rainfall observed by the CML."
- *To the discussion:* "Our results indicate that MLP_RA provide rainfall estimates that are more continuous, and more consistent over time, compared to the more intermittent estimates generated by MLP_GA (see for instance Fig. 1 time-step 06:00). This could come from the fact that the rain gauges have a 1-minute resolution while the weather radar has a 5-minute resolution, making the radar rainy periods more continuous. Another explanation could be that at low rainfall rates, the rain gauge will not record any rainfall before the droplets have been transported to the scale, making the period seem more intermittent than it actually is. Further, while the rain gauges measure point rainfall close to the CML, the weather radar measures average rainfall along the CML. This path averaging blurs the rainy periods, making the rainy period more continuous with fewer intermittent breaks. An interesting finding is that even though the rain gauges do not represent the average rainfall along the CML, MLP_GA is able to capture more of the underlying intermittency as compared to MLP_RA. This is also reflected in the neural network configuration where the MLP_GA benefits from a more complex network architecture as compared to MLP_RA."

We agree that the aggregation of the CML signal into the MLP could be more clearly explained. We have changed paragraph 65 so that it reads:

- "The MLPs job is to classify a time step in the CML time series as either rainy or dry. It does this by analyzing the signal loss from the surrounding 40 time steps. In essence, the MLP acts like a sliding window, moving across 40 time steps at a time, and determining whether each centered time step is rainy or dry."

The rain gauge data used in this study are provided by DWD, where the minimum resolution is 0,01mm in the data. In the data availability section we have provided a link to where this data can be obtained. The rain gauges used are of type rain[e] weighing precipitation sensor. More information on this rain gauge can be found here:
https://www.lambrecht.net/en/products/precipitation/weighing-precipitation-sensor-rain-e

2) I could not understand what do you want to "classify"? Do you want to distinguish rain or no-rain of instantaneous CML signal, or you want to adjust the CML signals to the precipitation intensity? Also, what does "feedforward" mean? Do you want to propose the new methods or just demonstrate how the MLP works? Please clarify the target (objects) of this technical note.

In this study we want to distinguish rain from no-rain on a higher temporal resolution than what is done by existing methods. The modifications done to the introduction (in response to comment 1) now describe the purpose more clearly.

To further answer comment 2) we have modified paragraph 45 so that the study's goals are more explicitly stated:

- "In this study, we present two methods for detecting rainy time steps in CML time series data. The goal of both methods is to detect rainy time steps in the time series of a CML where the signal attenuation is provided every 1 minute. This is done with a higher temporal resolution compared to existing methods so that short dry spells during rainy periods can be identified. One method is trained on radar reference data and the other method is trained on rain gauge reference data. Both methods are tested against rain gauge and radar data, highlighting their difference. We also examine the performance of the developed methods in comparison to existing approaches, aiming to gain a clearer understanding of the differences between the two alternative methods."

The word feedforward is a very common term in machine learning. We have chosen to keep it.

3) It is concluded that MLPs performed "better" than existing methods. Some statistics showed differences between your products and sigma80/CNN products, but comparisons on long-term sequence failed to estimate the variations (L184, Fig. 5). Please show the clear reasons of why your work is "better", and explain them in the conclusion. You propose the benefit of higher temporal estimates (L6-7), however it looks like failing filter out the short-term noise in Fig.5. Also, if you proud of the higher temporal product shorter than one hour, it is better to scale up the Figure 3 with one-minuets interval.

The MLPs perform better than existing methods when compared to 1-minute and 5-minute reference data. This is clearly shown in Figure 2. Figure 3 and 4 shows why this is the case: In short, the sigma80 and the CNN estimates much longer rainy periods, possibly because both of them were developed to predict rainfall using references with hourly resolution. The reason we use the MCC as a performance metric is explained in paragraph 85. The MLPs predict shorter wet periods that better reflect the intermittency of the references that record rainfall with a higher resolution. To better highlight this we have modified parts of the conclusion so that it reads:

- "In this technical note, we introduced two simple feedforward neural networks (MLPs) trained to detect rainy time steps in signal attenuation data from commercial microwave links (CMLs). The MLPs are trained and tested using reference data from rain gauges (MLP_GA) with a temporal resolution of 1 minute and gauge-adjusted radar (MLP_RA) with a temporal resolution of 5 minutes. Whereas existing methods tend to estimate longer continuous rainy periods, the MLPs estimate shorter rainy periods that more closely resemble the intermittent rainfall patterns that are observed by the rain gauges and weather radar. The performance of the MLPs is evaluated by comparing the MLPs estimates with estimates produced by two existing methods using the Matthews correlation coefficient. Our results show that the MLPs outperform existing methods in almost all cases."

Fig 3) clearly shows that the CNN and sigma80 method estimate longer rainy periods compared to the MLPs. Zooming in to a smaller time interval would leave out important details, like CNN estimating a very long rainy period. For assessing the methods' performance we think Fig 2) is very useful, as plotting the full time series of all links is unfeasible. Fig 3, 4 and 5 provides insight into where the MLPs excel and where they fail. We have chosen to keep Fig 3) as it is because it highlights important details for the discussion.

Fig. 5 shows a time series with a noisy CML. Normally, when deriving rainfall rates from CMLs, noisy CMLs are just completely removed from the dataset. The reason we included this was to show the limitations of the MLP and current rainfall detection method. To better highlight this we have:

- Rephrased paragraph 150 to read "To illustrate how the MLPs perform in comparison to the CNN and sigma_80 method, we have selected two events where the MLPs outperform the reference methods (Fig. 3 and Fig.4) and one event where the MLP performs less well (Fig. 5). The figures show the CML signal loss as a function of time as well as the estimated rainy periods for all methods and the ground truth. We also plot the confusion matrix and the corresponding MCC score for each method using the rain gauge as a reference."
- Added the following to the discussion: "Both MLPs were trained using the 26 CML-reference pairs that showed the highest MCC estimated using the sigma_80

method. This can be thought of as a pre-processing step, where the goal was to ensure training data with a good match between the reference and the CML. In our case this was important for making the MLPs converge to approximately the same weights every time we trained the model. These particular pairs might, since they by selection have a good correlation with their reference, also contain little or no noise. Thus, the MLP training datasets might lack exposure to noisy CML time series, and as a consequence, the MLPs might not very well handle noisy periods. On the other hand, from Fig. 2 we know that the MLPs still outperform the sigma_80 and CNN method on the 843 CMLs used in the test dataset, which was not subject to any noise filtering, suggesting that the MLPs at least to some extent can handle noise. Moreover, very noisy CMLs are typically handled using pre-processing methods such as filtering out CMLs with strong diurnal cycles or plateaus such as done in Graf2020 and Blettner2023."

Specific comments

Title: Not clear the meaning of "feedforward", "wet and dry period","

- As the term feedforward is quite well known in machine learning we have chosen to keep it. Regarding the wet and dry period we have used the term "rainy" instead.

L9 "signal attenuation" of what?

- To make the sentence clearer we have changed it to read: "Commercial microwave links (CMLs) are radio links between telecommunication towers. By exploiting the relation between CML signal attenuation and rainfall intensity, it is possible to estimate the average rainfall intensity along the CML"

L17 more and more "available" mean data are becoming open?

- yes, many of the now open datasets use instantaneous measurements,

L19 What is the "wet period", "Based on this,"?

- We have changed the sentence to read: "The CML signal experiences fluctuation during rain events. Based on this, a simple method for rain event detection was developed by schleiss2010."

L20 You frequently use "prediction", but this term is for the future estimation. Better to reconsider the usage as "estimate" such as in L175.

- We agree on this and suggest using the word "estimate" throughout the paper.

L32 "hourly reference data" means "hourly precipitation data"? Is this gauge data? No study using disdrometer?

- We agree that is a bit unclear and have rephrased to: "Although several of the mentioned approaches classify rainfall on a high temporal resolution, all large studies using instantaneous sampled CML data have been evaluated using hourly reference data."

L38 What is "rainy time step"? Non-liquid precipitation means solid precipitation such as snow fall?

- We propose to introduce the term rainy time steps earlier in the introduction, see suggestion above.
- Non-liquid precipitation could be snow, yes.

L40-41 I could not understand the meaning.

- We propose to change the sentence to read: "Further, a drawback of estimating too-long rainy periods is that some of the estimated rainy time steps could contain non-liquid precipitation. Because dry snow induces a very low signal attenuation, these time steps appear as dry in the CML time series. Thus, correctly estimating rainy time steps is important because CML time steps that indicate no precipitation could contain dry snow."

L43 Object of this note is to describe the methods? Please clarify the object here according to the conclusion.

- We have changed the objective to read: "In this study, we present two methods for detecting rainy time steps in CML time series data. The goal of both methods is to detect rainy time steps in the time series of a CML where the signal attenuation is provided every 1 minute. This is done with a higher temporal resolution compared to existing methods so that short dry spells during rainy periods can be identified. One method is trained on radar reference data and the other method is trained on rain gauge reference data. Both methods are tested against rain gauge and radar data, highlighting their differences. We also examine the performance of the developed methods in comparison to existing approaches, aiming to gain a clearer understanding of the differences between the two alternative methods."

2.1 Data: Please clarify the detailed network structure of the CML, such as general distance, distribution, and what is the "near the CML"? Which kind of rain gauge is used? How the instruments could get 1-minuet/0.01mm resolution? Is this optical sensor? 5 minuets resolution of DWD stans for every 5 minuet interval or 5 minuet average?

- The distance to the nearest rain gauge is explained later in the methods chapter.
- We suggest adding the following sentence to paragraph 60, section 2.1: "Our study focused on CML-rain gauge pairs located closer to each other than 5 km. This resulted in 882 CMLs where the CML lengths ranged from 0.3km to 22.9km with 90

percent of the CMLs being longer than 2.4km. The CML frequencies ranged between 7 GHz to 40 GHz, with most CMLs having a frequency above 15 GHz. Even though there are many CMLs in our dataset, we only have 429 unique rain gauges serving as references. This means that some CMLs use the same rain gauge for reference."

- The rain gauge data used in this study are provided by DWD, where the minimum resolution is 0,01mm in the data. In the data availability section we have provided a link to where this data can be obtained. To our knowledge the rain gauges used are of type rain[e] weighing precipitation sensor. More information on this rain gauge can be found here: https://www.lambrecht.net/en/products/precipitation/weighing-precipitation-sensor-rain-e

L60 "wet" may mean the existence of precipitation record. Then, there must be two kind of "wet periods" such as gauge based and radar based? Both periods were defined by the same threshold as 0.01mm per a minuet?

- We have changed the word "wet" to "rainy" to make it clearer. We have also added an explanation for the distinctions between a rainy and dry period in the introduction as well as some discussions regarding the approximation. See comments above

2.3-2.5 Some of this part are separated only by the paragraph. Please reconsider to combine the sub-section into one section composed by the story only by paragraphs.

- we think the current structure makes it easier for the reader to navigate the different concepts

L81 As the comparison of your methods to two previous method is the key, please explain in detail about two reference methods (Sigma80 and CNN).

- They are already explained in the introduction so we would prefer not to repeat them, as we try to keep this technical note as short as possible.

L112-119 This part describe the study direction, and better to move before. "intermittent rainfall" is your focus, so better to explain what it is (how did you detect).

- Good point! We have made changes to the introduction to better answer this. L112-L119 introduces the undersampling strategy.

L122 Delete "given optimal". Such type existed in may places.

- We have deleted the duplicate occurrence of "given optimal".

L126 What is "spatial difference"?

- We have rephrased to: "This could be because the rain gauges can be located up to 5km away from the CMLs, causing errors related to spatial variability"

L131 You insist "more consistently" from which part of the figure? Need more polite explanation to the readers.

- We have deleted the sentence as the following sentences fully explain the findings.

L150- I could not understand the process making Fig. 3-5. Are they case studies in different time series, or some kind of composite with different time scale? Better to divide upper and lower part as Fig. 3a and 3b, but the lower parts are not explained fully in the contents. White areas are dry periods?

- Figures 3-5 can be thought of as case studies. We have changed paragraph 150 to read: "To illustrate how the MLPs perform in comparison to the CNN and sigma_80 method, we have selected two events where the MLPs outperform the reference methods (Fig. 3 and Fig.4) and one event where the MLP performs less well (Fig. 5). The figures show the CML signal loss as a function of time as well as the estimated rainy periods for all methods and the ground truth. We also plot the confusion matrix and the corresponding MCC score for each method using the rain gauge as a reference."
- We have added a) and b) to the figures to make them easier to navigate.
- We also changed the term "wet" in the legends to "rainy".

L155 Why "nicely"? I can not catch up which part of the figure could correspond to your results. Same asn L161 "Further ,,, rise", etc.

- We suggest to rephrase the sentence to read: "has a relatively constant baseline outside the rainy time steps".
- To better highlight where the tl rises we have added: "see for instance time step 01:00".

L161 "Further, ,,rise." I could not identify those tendency in Fig. 3. Need marks or specify with times.

- To better highlight where the tl rises we have added: "see for instance time step 01:00".

Fig.3 upper: Clear characteristics on Fig.3, that I could identify, were such as 1) CNN is similar to GA and Sigma80 is similar to RA, 2) MPL-GA and MPL-RA estimated longer no-precipitation periods as RA, 3) MPL-GA and MPL-RA looks mostly similar with a small difference around 6:00. If the MLP studied the RA and GA observations respectively, why the MLP-GA and MLP-RA are so similar? Your focuses may be on more small scale, but I could

not figure out the importance of your argument. The graph of GA should put above the RA to adjust the order of MPL-GA and MPL-RA.

- Fig3 shows a test and thus we do not expect that MLP_GA should necessarily estimate the same wet periods as observed by the rain gauge. We do expect MLP_GA to learn the signal loss pattern of the CML that best explains the rainfall pattern at the rain gauge and the MLP_RA learns the signal loss pattern of the CML that best explains the rainfall patterns observed by the weather radar. The reason we included both the MLP_RA and MLP_GA was because they seem to estimate slightly different rainy periods compared to each other. Both outperforms existing methods.
- We agree to change the order of MLP-GA and MLP-RA as that makes the comparison more logical. See new figure.

L168 I could not identify the wet starting point.

- We have rephrased to: "estimates a rainy period starting at 12:00, shortly before MLP_GA estimates a wet period."

L170 I could not understand the description of "non of the methods ,,". What is the "reference period"?

- We have rephrased to: "In this case, none of the CML rainfall detection methods can accurately estimate the radar or rain gauge reference rainy periods."

L173 "However, " I could not understand what you mean.

- We suggest deleting the sentence as it is fully covered by the previous sentences.

L178-182 This part should move to the paragraph in L166.

- We have keep this part as it is and rephrased the paragraphs in L166 to better guide the reader in reading through the figures using time marks such as "estimates a rainy period starting at 12:00, shortly before MLP_GA estimates a wet period."

Fig.5 This is the result of longest record with few precipitation periods in RA GA where MPL reproduced some intermittent periods of rains similar to sigma80. The author mentioned "erratic signals" in L184, and attributed by a noisy CML signal. It means the MLP even could not filter out one noise on the CML? Then why you can conclude "better (L3)" performance?

- From Figure 2 we can see that the MLPs outperform the existing methods. Please note that focusing on these specific time steps of Figure 3-5 was intentional and that by choosing other time steps we would get different results. This particular case was chosen because it shows an edge case where the MLPs fail and we believe that the reader should be aware of this.

- We acknowledge that part of the job of the MLPs is to filter out noise in the CML time series. However, this CML in particular is very noisy and would most likely be disregarded by quality control routines.
- Future developments into wet and dry detection could definitely seek to use for instance deep learning methods to identify fluctuations due to other-than-rain factors in CML time series. This is however far from the scope of this work where we have used very simple network architectures.
- We have addressed this by adding the following:
  - To paragraph 150: "To illustrate how the MLPs perform in comparison to the CNN and sigma_80 method, we have selected two events where the MLPs outperform the reference methods (Fig. 3 and Fig.4) and one event where the MLP performs less well (Fig. 5). The figures show the CML signal loss as a function of time as well as the estimated rainy periods for all methods and the ground truth. We also plot the confusion matrix and the corresponding MCC score for each method using the rain gauge as a reference."

L190-195 Future issues should be mentioned in the conclusion.

- We think that L190-195 fits best into the discussion part.